# The Dynamics and Plasticity of Epigenetics in Diabetic Kidney Disease: Therapeutic Applications Vis-à-Vis

**DOI:** 10.3390/ijms23020843

**Published:** 2022-01-13

**Authors:** Feng-Chih Kuo, Chia-Ter Chao, Shih-Hua Lin

**Affiliations:** 1National Defense Medical Center, Department of Internal Medicine, Division of Endocrinology and Metabolism, Tri-Service General Hospital, Taipei 114, Taiwan; shoummie@hotmail.com; 2Department of Internal Medicine, Nephrology Division, National Taiwan University Hospital, Taipei 100, Taiwan; b88401084@gmail.com; 3Graduate Institute of Toxicology, National Taiwan University College of Medicine, Taipei 100, Taiwan; 4Department of Internal Medicine, Nephrology Division, National Taiwan University College of Medicine, Taipei 100, Taiwan; 5National Defense Medical Center, Graduate Institute of Medical Sciences, Taipei 114, Taiwan; 6National Defense Medical Center, Department of Internal Medicine, Nephrology Division, Taipei 114, Taiwan

**Keywords:** diabetic kidney disease, epigenetics, DNA methylation, histone modifications, noncoding RNAs

## Abstract

Chronic kidney disease (CKD) refers to the phenomenon of progressive decline in the glomerular filtration rate accompanied by adverse consequences, including fluid retention, electrolyte imbalance, and an increased cardiovascular risk compared to those with normal renal function. The triggers for the irreversible renal function deterioration are multifactorial, and diabetes mellitus serves as a major contributor to the development of CKD, namely diabetic kidney disease (DKD). Recently, epigenetic dysregulation emerged as a pivotal player steering the progression of DKD, partly resulting from hyperglycemia-associated metabolic disturbances, rising oxidative stress, and/or uncontrolled inflammation. In this review, we describe the major epigenetic molecular mechanisms, followed by summarizing current understandings of the epigenetic alterations pertaining to DKD. We highlight the epigenetic regulatory processes involved in several crucial renal cell types: Mesangial cells, podocytes, tubular epithelia, and glomerular endothelial cells. Finally, we highlight epigenetic biomarkers and related therapeutic candidates that hold promising potential for the early detection of DKD and the amelioration of its progression.

## 1. Introduction

Chronic kidney disease (CKD) is an instrumental risk factor for cardiovascular morbidities and has become the 12th leading cause of global death [1]. CKD confers a substantial economic burden on the existing healthcare system, since CKD incurs high medical costs, especially involving those with end-stage renal disease (ESRD) requiring long-term dialysis [2]. Overall, around 10% to 20% the global population has CKD, and the prevalence of CKD is still on the rise, particularly in developed countries [3]. The clinical presentations of CKD include proteinuria and a progressive loss of kidney function as measured by lowered estimated glomerular filtration rates (eGFRs). The ultrastructural alterations in kidneys of patients with CKD consist of glomerular or renal tubular injuries manifesting in podocyte foot process effacement, mesangial hypertrophy, extracellular matrix (ECM) accumulation, tubular epithelial-to-mesenchymal transition (EMT), and interstitial infiltration of inflammatory cells [4]. These pathological processes eventually induce irreversible renal fibrosis and clinically present as ESRD, with which patients exhaust most of their functional glomeruli. Therefore, timely discovery of the molecular pathogenesis of CKD with the administration of appropriate therapies can be crucial for retarding kidney dysfunction in patients with CKD.

To prevent CKD progression, the optimized control of hyperglycemia, hypertension, and dyslipidemia is fundamental. The avoidance of repeated acute kidney injury (AKI), the use of reno-protective medications such as renin-angiotensin-aldosterone system blockers in patients with hypertension or sodium glucose cotransporter 2 inhibitors in patients with diabetes mellitus (DM), and the prompt correction of complications such as anemia, metabolic acidosis, and electrolyte imbalances, are important approaches for retarding CKD and reducing morbidity/mortality [5]. Other factors such as genetic susceptibility and epigenetic dysregulations also play a role in the progression of CKD [6].

Genome-wide association studies (GWAS) involving patients with CKD have identified > 250 genetic loci that correlated significantly with kidney function-related traits [7,8,9]. Recently, studies using the epigenome-wide next-generation sequencing and whole transcriptome analysis have pinpointed epigenetic alterations as important mechanisms underlying the interactions between the environment and the genome, and these alterations modify an individual’s vulnerability and his/her risk for CKD development [10,11]. Since diabetes emerges as the main etiology of CKD, we review the existing literature regarding the epigenetic alterations inherent to diabetic kidney disease (DKD) and summarize cell-specific epigenetic mechanisms for major kidney cell types. Findings elaborated in this article are expected to facilitate individualized therapies based on the predominant cells involved in DKD pathology.

## 2. Epigenetic Machineries: An Introduction

Epigenetic regulation is defined as any process that influences transcription without changing DNA sequences. Three major epigenetic regulatory mechanisms have been delineated, including DNA methylation, chromatin modifications, and noncoding RNAs. They also interact with each other to orchestrate the complex epigenetic landscape associated with the regulation of nuclear 3D architectures, chromatin condensation, and DNA accessibility of enhancers, repressors, or promoter regions. Epigenetic regulation is dynamically influenced by an individual’s surrounding environmental and can be inherited through cell divisions or between generations through the transmission of epigenetic markers to offspring [12,13]. The process of epigenetic regulation is also reversible or adjustable [14,15]. Epigenetic dysregulation has been identified as a major driving force for multiple diseases. Furthermore, epigenome-targeted therapies have been approved by the United States Food and Drug Association (FDA) for treating myelodysplastic syndrome and leukemia [14,15]. Recent studies have identified crucial epigenetic markers capable of impeding CKD progression. In the following sections, we will briefly describe the major epigenetic machineries.

### 2.1. DNA Methylation

DNA nucleotides consist of three subunits, the nucleobase, the five-carbon deoxyribose, and the phosphate group. There are four different DNA nucleobases: Guanine (G), adenine (A), cytosine (C), and thymine (T). CpG sites are DNA regions where a cytosine nucleotide is followed by a guanine nucleotide along the 5’ to 3’ direction, while CpG islands refer to regions containing a high frequency of CpG sites. The cytosines of CpG sites can be methylated, forming 5-methylcytosines (5 mCs), and around 80–90% CpG sites are methylated [16]. In the mammalian genome, approximately 70% of proximal promoters contain CpG islands near the transcription start sites [17,18], and the presence of multiple methylated CpG sites in the promoter region suppresses gene expressions [19]. The promoter CpG islands of housekeeping genes are mostly hypomethylated, leading to a constitutive transcription. The maintenance of methylation is achieved by DNA methyltransferases 1 (DNMT1) via its strong preference for methylating CpG sites on the hemi-methylated DNA to counteract the dilutive effect of mitosis [20,21,22]. On the other hand, DNMT3A and DNMT3B generate de novo methylation, modifying methylation patterns [23,24,25]. The reversal for DNA methylation is achieved by changing 5 mC to 5-hydroxy-methyl-cytosine (5 hmC) by Ten-Eleven Translocation (TET) [26] (Figure 1A). Through concerted actions between DNMT and TET, the global DNA methylation status can be stably maintained and dynamically modulated during cellular proliferation, differentiation, and the stressed condition [27].

### 2.2. Histone Modifications

Histones are basic amino acid-abundant proteins (lysine, abbreviation K, and arginine, abbreviation R) that wrap the double-helix DNA to create nucleosomes [28], which contain 4 ‘core’ histones (H2A, H2B, H3, and H4). The H2A-H2B dimers and the H3-H4 tetramers form the histone octamers [29]. Nucleosomes are further wrapped into 30-nanometer fibers and tightly packed as chromatin. The interactions between histones, DNA, and nuclear proteins are altered via post-translational modifications (PTMs) involving lysine or arginine residues. Histone modifications are important epigenetic machineries influencing gene expressions, DNA repair, and chromatin condensation [30]. The major PTMs of histones include acetylation, methylation, phosphorylation, ubiquitylation, and SUMOylation [31] executed by histone modifiers such as writers (histone acetyltransferases and methyltransferases), erasers (deacetylases and demethylases), and readers (bromodomain-containing proteins (BRDs) such as BRD4) [10] (Figure 1B).

Histone acetyltransferases such as p300 and CREB-binding protein (CBP) catalyze histone acetylation to unwind chromatin and increase the accessibility of transcription factors or co-factors to promoters/enhancers to modulate gene expressions. Histone deacetylases (HDACs) and sirtuins remove acetylation marks of histones and act as co-repressors [32,33]. Histone methyltransferases or demethylases have variable functional roles in activating or repressing transcription through inducing different residue methylation and modification [34,35]. For example, lysine methylations at H3K4, H3K36, and H3K79 are associated with upregulation and enriched at transcriptionally active regions. On the other hand, methylation at H3K9, H3K27, and H4K20 repress gene repressions and are enriched at repressed regions [34,36]. The epigenetic readers of BRDs recognize acetylated lysine residues and regulate protein–protein interactions to activate transcription [37]. These histone modifiers jointly modulate transcription, but their interactions are still incompletely understood.

### 2.3. Noncoding RNAs

Noncoding RNAs are RNA molecules not translated into proteins. They can be divided into small and long noncoding RNAs based on total nucleotide counts. Small noncoding RNAs have fewer than 200 nucleotides and include three classes, microRNAs (miRNAs), small interfering RNAs (siRNAs), and piwi-interacting RNAs (piRNAs). MiRNAs are small single-strand non-coding RNAs (about 22 nucleotides) that function to silence target messenger RNAs (mRNAs) via base-pairing with complementary sequences [38] (Figure 1C). siRNAs are double-strand non-coding RNAs (about 20–24 base pairs) that work similarly to miRNAs via degrading mRNAs [39]. piRNAs have around 26–31 nucleotides and act by forming piRNA complexes with piwi-subfamily Argonaute proteins [40,41].

Long non-coding RNAs (lncRNAs) have more than 200 nucleotides with secondary or tertiary structures, serving as scaffolds for interactions between RNAs, DNAs, and proteins. LncRNAs are expressed in relatively low levels with tissue specificity and exhibit multiple regulatory mechanisms including sponging miRNAs or inducing histone modifications [42] (Figure 1C). The functions of many lncRNAs are still uncertain and require further investigation.

## 3. DKD: An Overview

Type 1 and type 2 DM are the main etiologies for CKD development. Approximately 45% of cases with ESRD result from DKD [43], and 20–40% of patients with type 2 DM have nephropathy depending upon geographic areas [44]. Hyperglycemia is an important form of metabolic stress capable of modifying epigenetic processes [45]. Many studies evaluate epigenetic modifications in DKD; in the following section, we summarize reports investigating the pathogenic roles of epigenetic regulation associated with DKD.

### 3.1. DNA Methylation in DKD

Prior studies used whole blood- or mononuclear cell-derived DNAs for performing epigenome-wide association studies (EWAS). Bell et al. conducted a case-control study of 192 Irish type 1 diabetic patients to compare DNA methylomes between individuals with and without nephropathy. They identified 19 CpG sites associated with the time to DKD development [46]. Similarly, studies involving patients with type 1 DM used the Illumine Infinium HumanMethylation 450 array and identified differentially methylated genes, especially those altering mitochondrial function [47]. In the Diabetes Control and Complications Trial (DCCT) and the Epidemiology of Diabetes Interventions and Complications (EDIC) studies, investigators used whole blood- and monocytes-derived DNA for comparing global methylomes between those with and without diabetic complications [48]. They identified 12 differentially methylated loci common to both whole blood- and monocyte-derived DNA, including hypomethylation of *TXNIP* (thioredoxin-interacting protein involved in oxidative stress and apoptosis) [49]. Their results provide strong evidence that hyperglycemic-mediated DNA methylation creates metabolic memory related to the DKD pathogenesis.

EWAS using whole blood-derived DNA has also been performed in patients with type 2 diabetes. Several CpG hypomethylation sites were found to correlate with genes involved in DKD [50]. After combining methylomes, transcriptomes, and single nucleotide polymorphisms (SNPs) from 500 patients with DKD from the Chronic Renal Insufficiency Cohort, the authors found 40 differentially methylated and expressed loci prioritized, exhibiting associations with DKD phenotypes, and these loci were functionally enriched for inflammation, apoptosis, and complement activation [51]. Another large study analyzing methylomes from the Atherosclerosis Risk in Communities cohort and the Framingham Heart study cohort revealed 19 CpG sites associated with eGFR, among which five were associated with biopsy-proven renal fibrosis; these findings indicate that changes in DNA methylation occurred in the kidney cortex [52]. Similarly, aberrant DNA methylation was also demonstrable in proximal tubular epithelial cells collected from type 2 diabetic animals, and these genes were functionally enriched for mitochondrial biogenesis [53]. Collectively, hyperglycemia-induced DNA methylation changes are both evident in patients with DKD associated with type 1 and type 2 DM and affect not only circulating blood cells but also kidney cells.

Changes in DNMT1 levels in whole blood or mononuclear cells from patients with DKD also participate in its pathogenesis. DNMT1 levels were found to increase in the peripheral blood mononuclear cells (PBMCs) of patients with DKD and a higher inflammation severity [54]. Another study also indicated that DNMT1 activity increased during DKD [55]. A higher DNA methylation ratio of let-7a-3 promoter with increased expressions of *UHRF1* and *DNMT1* was found in those with DKD [55], suggesting that loss of let-7a-3 might restore the expression of its target, UHRF1, which enhanced the binding of DNMT1 to hemi-methylated DNAs [56]. The increased DNMT1 levels in circulating blood cells might influence multiple organs or systems, since DNMT1 aggravated retinal damages in diabetic mice [57].

The correlation between gene methylation levels and clinical DKD markers such as albuminuria, serum homocysteine [58], and insulin-like growth factor binding protein-1 (IGFBP-1) [59] also warrants discussion. Higher albuminuria levels were associated with hypomethylation of *TIMP-2* and *AKR1B1* [60]. Increased serum homocysteine was also observed in patients with DKD, and the methylene tetrahydrofolate reductase (*MTHFR*) gene, which was involved in homocysteine metabolism, had higher promoter methylation levels [61]. Patients with type 1 DM and DKD had higher circulating IGFBP-1 and lower *IGFBP1* methylation levels [62], suggesting that IGFBP-1 participated in DKD development. In addition, the connective tissue growth factor (CTGF) is an inducer for extracellular matrix accumulation and correlates with DKD pathogenesis [63]. Interestingly, lower *CTGF* DNA methylation levels and increased CTGF protein were associated with albuminuria and eGFR decline in patients with DKD [64], compatible with results from cellular studies showing that high glucose-treated mesangial cells exhibited hypomethylation of the *CTGF* promoter and increased CTGF expressions [65]. Moreover, the RAS protein activator-like 1 (RASAL1) encoded by *Rasal1* inhibits renal fibrosis by switching off RAS activity [66]. Using kidneys from unilateral ureteral obstruction (UUO) and DKD mice and primary renal fibroblasts, hypermethylation of the *Rasal1* promoter contributed to fibroblast activation and kidney fibrosis progression [67]. Overall, these results support the fact that multiple signaling pathways are dynamically regulated by DNA methylation, contributing to DKD progression. We summarize these results in Table 1.

### 3.2. Histone Modifications in DKD

Hyperglycemia-mediated epigenetic processes and metabolic memory are not limited to DNA methylation. A case-control study aimed to analyze the promoter enrichment for H3 lysine-9 acetylation (H3K9ac), H3 lysine-4 trimethylation (H3K4me3), and H3K9me2 in monocytes and lymphocytes from EDIC and DCCT participants [68]. Notably, monocytes from cases presenting with increased H3K9ac enrichment (active chromatin mark) in promotor regions were significantly associated with mean HbA1c levels. More than 15 genes were linked to the top 38 hyperacetylated promoters, associated with the nuclear factor-kB (NF-κB) inflammatory pathway and diabetes complications. These results indicate that the active histone posttranslational modification H3K9ac is an important epigenetic mark for metabolic memory in patients with type 1 diabetes. Several studies also pointed out that histone acetyltransferases (such as p300 and CBP) and deacetylases (such as HDACs and SIRT1) participated in the epigenetic regulation of DKD. Using a diabetic mice model and transforming growth factor (TGF)-β1-treated mesangial cell lines, p300- and CBP-mediated H3 acetylation (such as H3K9/14 acetylation) were enriched at *PAI-1* and *p21* promoters [69], and the acetylation of transcription factor Ets-1 was induced by p300 [70]. Ets-1 subsequently sustained the expressions of miR-192 [71]. Plasminogen activator inhibitor-1 (PAI-1), p21, and miR-192 were all involved in glomerular hypertrophy and extracellular matrix accumulation [72,73,74], contributing to DKD pathogenesis.

Regarding histone deacetylases, increased HDAC4 and HDAC9 were found in the renal biopsy specimens of patients with DKD and diabetic animals [75,76]. Their roles in high glucose-induced reactive oxygen species (ROS) generation, podocyte apoptosis, and inflammation were also demonstrated [75,76]. Downregulation of Sirt1 was observed in the proximal tubules (PTs) of diabetic mice, and the protective roles of PT Sirt1 in glomerular podocytes were further disclosed using the PT-specific SIRT1 transgenic and Sirt1 knockout mice. These results highlighted the intricate crosstalk between PT cells and podocytes [77]. On the other hand, apelin-13 could attenuate the development of DKD via inhibiting histone hyperacetylation and suppressing inflammatory factors [78]. In aggregate, these results suggest that histone acetyltransferases and deacetylases actively engage in the pathogenesis of DKD.

Studies on histone methyltransferases (SET7/9, SUV39H1, and EZH2) reveal their involvement in DKD progression. Reports of TGF-β1 or high glucose-induced extracellular matrix (ECM) gene expressions in rat mesangial cells revealed that SET7/9 (a lysine methyltransferase for H3K4) and active chromatin marks (H3K4me1, H3K4me2, and H3K4me3) were increased and recruited to their promoter regions [79]. Moreover, in the kidneys of type 2 diabetic mice, an increase in X-box binding protein 1 (XBP1s), an endoplasmic reticulum (ER) stress-inducible transcription factor, promoted the expression of SET7/9 with augmented recruitments of SET7/9 and H3K4me1 on the promoters of monocyte chemoattractant protein-1 (MCP-1) [80]. SUV39H1 catalyzing the trimethylation of histone H3 lysine 9 (H3K9me3) also correlated with mesangial dysfunction or tubular inflammation in the setting of DKD. Downregulation of SUV39H1 reduced H3K9me3 recruitment on the promoters of fibronectin and p21 in high glucose-treated mesangial cells, increasing their expressions and stimulating mesangial hypertrophy [81]. Likewise, greater and prolonged glucose stimulation of human PT cells suppressed SUV39H1 and H3K9me3 with increased expressions of IL-6 and MCP-1 [82]. SET7/9 and SUV39H1 expressions could be modulated by H2AK119 and H2BK120 mono-ubiquitination in diabetic kidneys [83]. EZH2 catalyzing the trimethylation of histone H3 lysine 7 (H3K27me3) was shown to protect against DKD progression. Pharmacologic or genetic depletion of EZH2 in diabetic kidneys or high glucose cultured podocytes could decrease H3K27me3 and induce *TXNIP* expression, augmenting oxidative stress and proteinuria [84]. Downregulation of H3K27me3 was observed in podocytes from patients with DKD, and EZH2 knockdown in podocytes promoted its dedifferentiation while restoring Notch ligand Jagged-1 expression [85]. In mesangial cells and rodent DKD model, decreased enrichment of Ezh2 and H3K27me3 at the promoters of ECM or inflammatory genes were associated with mesangial dysfunction [86]. Taken together, the post-translational modifications of histones are heavily involved in the development of DKD. We summarize these findings in Table 2.

### 3.3. Noncoding RNAs in DKD

Noncoding RNAs including miRNAs and lncRNAs are widely investigated in DKD. Decreased expressions of miR-451 are related to increasing LMP7 and NF-κB expressions within PBMCs of DKD patients and in high glucose cultured mesangial cells [87]. The suppression of miR-146a induced inflammation in kidneys and macrophages of type 1 diabetic mice [88], while increased ErbB4 and Notch-1 signaling caused podocyte injuries [89]. Overexpression of miR-192 occurred in kidneys of type 1 diabetic mice and TGF-β-treated mesangial cells, enhancing ECM accumulation and fibrosis through pathways including p53, E-box repressors, and Smad-interacting protein 1 (SIP1) [72,90,91]. On the contrary, decreasing miR-192 in diabetic kidneys and high glucose-treated PT cells correlated with impaired renal function and TGF-β-induced epithelial-mesenchymal transition (EMT) [92]. MiR-29 family members, including miR-29a, 29b, and 29c, all play an important role in DKD pathogenesis. Downregulated miR-29a enhanced HDAC4 activity to induce nephrin deacetylation, leading to podocyte injury [93]. Lower miR-29b expression restored Sp1 expressions to stimulate TGF-β and NF-κB while increasing collagen matrix production by mesangial cells [94]. MiR-29b was also sponged by Erbb4-IR, an lncRNA, to promote fibrosis in both mesangial and tubular epithelia [95]. Elevation of miR-29c occurred in podocytes and correlated with ECM accumulation via suppressing Sprouty homolog 1 (Spry1) [96]. MiR-34 family members such as miR-34a and 34c were also associated with DKD, since the Notch signaling pathway was activated by decreasing miR-34a or miR-34c, causing podocyte apoptosis [97,98]. In kidneys from type 2 diabetes mice and mouse mesangial cells, decreasing lncRNA 1700020I14Rik was linked to the upregulation of miR-34a-5p/Sirt1/HIF-1α signaling and induced renal fibrosis [99]. Other miRNAs contribute to DKD progression via increasing ECM accumulation or promoting EMT. For example, miR-21, 200b/c, 135a, or 377 promoted ECM accumulation through regulating smad7 [100,101], Akt [102], transient receptor potential cation channel subfamily C member 1 (TRPC1) [103], or fibronectin expression [104]. Loss of miR-26a or let-7b/c/d/g/i altered ECM accumulation via changing CTGF [105], TGF-beta1 receptor 1 (TGFBR1) [106], or Lin28 [107]. Furthermore, downregulation of miR-130b, 200a, or 141 promoted EMT through Snail [108] or smad3 [109] signaling.

LncRNAs can interact with miRNAs to orchestrate their actions on DKD. For example, interactions of 1700020I14Rik-miR-34a-5p and Erbb4-IR-miR-29b are related to DKD, while several lncRNAs complementarily sponge their target siRNAs. During DKD, NR_033515 negatively regulated miR-743b-5p to promote proliferation and EMT of mesangial cells [110]; MALAT1 repressed miR-23c and modulated tubular epithelial pyroptosis [111], and the downregulation of LINC01619 followed by the restoration of miR-27a levels increased ER stress in podocytes [112]. Some lncRNAs are co-transcribed with miRNAs. For instance, lnc-MGC and a megacluster of miRNAs were coordinately expressed to induce mesangial hypertrophy and ECM accumulation in DKD models [113]. PVT1 and miR-1207-5p together contributed to glomerular ECM accumulation under hyperglycemia [114], and MIR503HR co-transcribed with miR-503-5p to exacerbate apoptosis in high glucose-treated tubular epithelia [115]. All in all, many miRNAs and lncRNAs play important roles in the pathogenesis of DKD and work intimately to form intricate regulatory systems. We summarize these findings in Table 3.

## 4. Kidney Cell-Centric View of Epigenetic Modifications

We further summarize regulatory actions exerted by three major epigenetic mechanisms on DKD. Studies using samples from patients with CKD or animal models of CKD without focusing on a specific type of CKD are not included here. To concisely depict the diverse epigenetic regulations from a cell-centric view, we schematically constructed diagrams to describe epigenetic modifications associated with mesangial cells (Figure 2), podocytes (Figure 3), and tubular epithelia (Figure 4). However, experimental studies measuring epigenetic marks from whole kidney samples without addressing specific kidney cells are not detailed in figures due to the lack of relevant evidence. A future approach may be able to leverage our summary results for devising target therapy for ameliorating DKD progression.

## 5. Epigenetic Modifications of Glomerular Endothelial Cells in DKD

DKD is a microvascular complication of diabetes, and several studies found that hyperglycemia-mediated epigenetic alternations promoted glomerular endothelial dysfunction. For example, the endothelial nitric oxide synthase (eNOS), a major enzyme in renal vasculature, has been shown to be regulated by histone modification [116]. The decreased NO production largely contributes to the endothelial dysfunction observed in DKD [117]. Indeed, using eNOS knockout diabetic mice as an animal model of advanced DKD for comparison with models with mild DKD and WT diabetes, Fu et al. disclosed that the differentially expressed genes in isolated glomerular endothelial cells were enriched functionally for angiogenesis and epigenetic regulation [118]. Another study investigating the role of long non-coding RNA metastasis-associated lung adenocarcinoma transcript 1 (MALAT1) in high glucose-induced glomerular endothelial dysfunction also showed that MALAT1 could epigenetically inhibit klotho transcription via recruiting methyltransferase G9a to increase H3K9me1 occupancy on the *klotho* promoter [119]. Moreover, culturing glomerular endothelial cells with conditional media from high glucose-treated podocytes could increase the phosphorylation of histone H3 serine 10 at the vascular cell adhesion molecule 1 (*VCAM1*) promoter, with increased protein productions [120]. These findings suggest that cells within glomeruli may interact with each other and contribute to DKD pathogenesis.

## 6. Epigenetic Biomarkers

Epigenetic markers, such as microRNAs, can be detected in the circulation and urine. Multiple studies have examined miRNA levels in patients’ blood and urine, aiming to identify useful biomarkers for early diagnosis and assessment of disease activity in DKD. The elevation of miR-192, 194, and 215 in urinary exosomes was considered as a biomarker for patients with type 2 DM and early DKD [121]. In patients with type 1 DM and incipient DKD, urinary exosomal miR-130a and 145 increased significantly while miR-155 and 424 drastically decreased [122]. In DKD, the reduction in circulating miR-126 was negatively correlated with albuminuria [123] and a higher risk for DM-related micro- and macro-complications [124], whereas the downregulation of urinary miR-2861, 1915-3p, and 4532 was associated with impaired renal function [125]. We summarize these findings in Table 4.

## 7. Epigenetic Machineries for Therapeutic Applications

Epigenome-targeting drugs including DNA methylation inhibitors, histone modifiers, and bromodomain inhibitors have been approved for clinical use or tested in oncology trials [126]. These drugs are also potential candidates for treating other diseases with epigenetic dysregulation, such as DKD. Indeed, as shown in Table 4, DNA demethylating agents (5-azacytidine or 5-aza-2’-deoxycytidine) attenuate diabetes-related podocyte injuries in animal models [127]. Bone morphogenic protein 7 (BMP7) administration can increase Tet3 expression and reverse the *Rasal1* promoter hypermethylation with the amelioration of renal fibrosis in UUO and DKD animal models [67]. The adipokine apelin-13 has been shown to attenuate DKD progression by inducing histone deacetylation [78]. Moreover, MS417 as a bromodomain and extraterminal (BET)-specific bromodomain inhibitor can attenuate proteinuria in DKD via blocking acetylation-mediated association of p65 and STAT3 with BET proteins [128].

Although DNA demethylation agents and histone modifiers show great promise for treating DKD, their broad and non-specific long-term effects are still of concern. Recently, the CRISPR-Cas9 system has been developed to perform site-specific epigenetic modifications [129,130]. Applying this technology to precisely and timely edit epigenetic elements may broaden the therapeutic armamentarium against DKD.

## 8. Conclusions

Multifactorial interventions to control hyperglycemia, hypertension, and dyslipidemia remain fundamental for ameliorating late complications in patients with DKD, and this approach has been proven in long-term multicenter randomized control trials [131]. Moreover, cardiovascular outcome trials of sodium-glucose co-transporter 2 inhibitors (SGLT2i) have been reported to beneficially influence cardiovascular and renal outcomes in patients with type 2 DM [132]. Therefore, a multifactorial approach, in addition to the use of nephroprotective agents, serves as a practical strategy for managing DKD. Epigenetic machineries undoubtedly play an active role in steering the initiation and progression of DKD. Our current understandings of the epigenetic dynamics during DKD progression remain fragmented. The interactions between different epigenetic processes enrich the complexity. Nonetheless, the increased availability of multi-omic approaches will provide us with more insight into epigenomic pathology in DKD. With the rising applicability of epigenome-directed editing technology, epigenetic therapies will likely be applicable in the future as a safe and effective way to treat DKD.

## Figures and Tables

**Figure 1 ijms-23-00843-f001:**
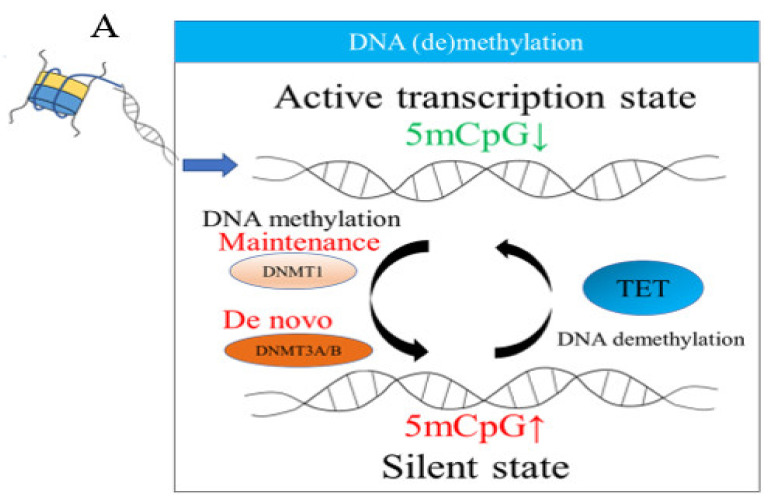
Schematic presentations of three major epigenetic regulations. (**A**) DNA (de)methylation; (**B**) histone modifications; and (**C**) noncoding RNAs. Denote: 5 mCpG: A methyl cytosine followed by guanine along the 5’ to 3’ direction, DNMT: DNA methyltransferases, TET: Ten-Eleven Translocation, CBP: CREB-binding protein, HDACs: Histone deacetylases, BRDs: Bromodomain-containing proteins, Ac: Acetylation, miRNAs: MicroRNAs, and LncRNAs: Long noncoding RNAs.

**Figure 2 ijms-23-00843-f002:**
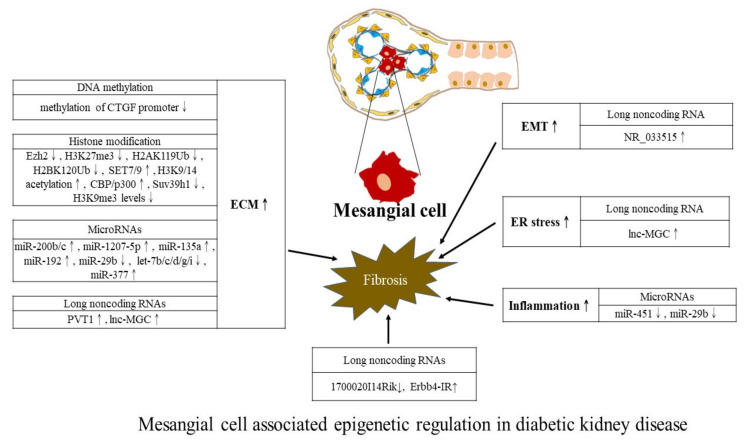
Mesangial cell associated epigenetic regulation in DKD. Denote: ECM: Extracellular matrix, EMT: Epithelial-to-mesenchymal transition, and ER: Endoplasmic reticulum. Symbols: ↑ refers to increase; ↓ refers to decrease.

**Figure 3 ijms-23-00843-f003:**
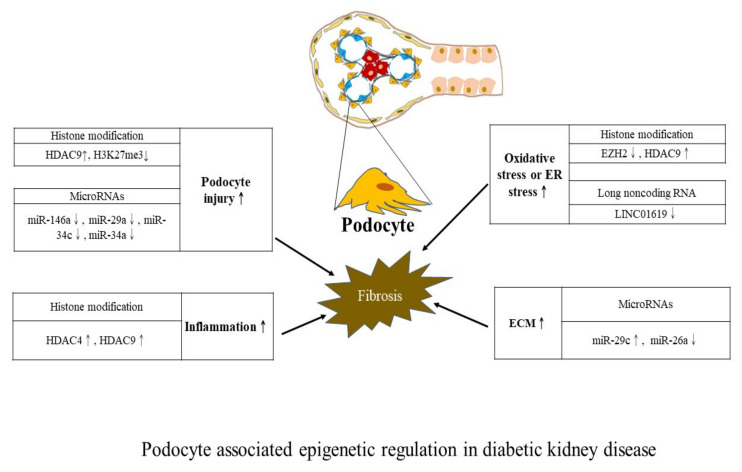
Podocyte-associated epigenetic regulation in DKD. Denote: ECM: Extracellular matrix and ER: Endoplasmic reticulum. Symbols: ↑ refers to increase; ↓ refers to decrease.

**Figure 4 ijms-23-00843-f004:**
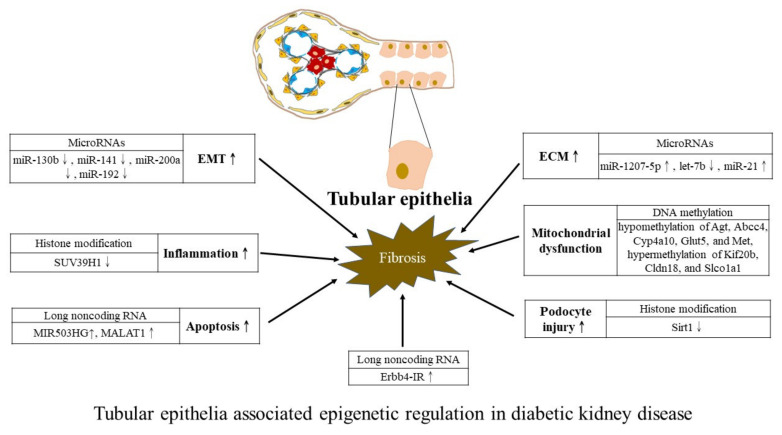
Tubular epithelia-associated epigenetic regulation in DKD. Denote: ECM: Extracellular matrix and EMT: Epithelial-to-mesenchymal transition. Symbols: ↑ refers to increase; ↓ refers to decrease.

**Table 1 ijms-23-00843-t001:** DNA methylation in diabetic kidney disease.

Study Design (Reference)	Main Cells or Tissue Samples	Epigenetic Changes (Mechanisms Involved)
A case-control study of 192 Irish T1D patients. Cases had T1D and nephropathy whereas controls had T1D without renal disease [46]	Whole blood	Methylation state of 19 CpG sites associated with risk of diabetic kidney disease (EWAS, time to diabetic kidney disease)
A case-control association study (*n* = 196 T1DM and diabetic kidney disease vs. *n* = 246 without renal disease) [47]	Whole blood	*PMPCB*, *TSFM,* and *AUH* with differential methylation at multiple CpG sites (EWAS, mitochondria dysfunction)
DNA from Pre-DM (*n* = 11) at baseline and at their transition to T2DM [50]	Whole blood	694 CpG sites hypomethylated and 174 CpG sites hypermethylated (EWAS, glucose/lipid metabolism, and inflammation)
Genome-wide methylome in 500 subjects with DKD from the Chronic Renal Insufficiency Cohort [51]	Whole blood	Prioritized 40 loci, methylation and gene-expression changes likely mediate the genotype effect on kidney disease development (EWAS, inflammation↑)
60 individuals, with 20 cases in the control, DM and DKD groups respectively [55]	Whole blood	Higher methylation ratio of the let-7a-3 promoter (UHRF1↑and DNMT1↑)
Two groups of patients based on albumin excretion as patients with (*n* = 69) and without DKD (*n* = 27) [60]	Whole blood	Hypomethylation of *TIMP-2* and *AKR1B1* genes (albuminuria↑)
24 cases of simple diabetes group; 34 cases of early DKD group; 27 cases of clinical DKD group; and 30 healthy controls [61]	Whole blood	Higher *MTHFR* promoter methylation in clinical diabetic kidney disease group (homocysteine↑)
778 Swedish individuals, including T1D patients with or without DKD and subjects with normal glucose tolerance [62]	Whole blood	DNA methylation levels in the *IGFBP1* gene↓ (circulating IGFBP-1↑)
Non-diabetes control (*n* = 29), diabetes without nephropathy (*n* = 37), and diabetes with nephropathy (*n* = 38) [64]	Whole blood	Lower *CTGF* methylation levels (ECM↑, albuminuria↑)
32 cases (conventional therapy with retinopathy or albuminuria) vs. 31 subjects (intensive therapy without complication), human monocytes [48]	Whole blood isolated at EDIC Study year 10 and monocytes during year 16–17.	12 differentially methylated loci were common in both whole blood and monocytes, including hypomethylation of *TXNIP* (EWAS, oxidative stress↑)
Mononuclear cells in DKD patients, diabetic mice, and cultured diabetic mononuclear cells [54]	Immune (mononuclear) cells	DNMT1↑ (inflammation↑)
Whole-blood DNA methylation of 2264 (586 DM) Atherosclerosis Risk in Communities and 2595 (394 DM) Framingham Heart Study participants [52]	Whole blood and renal biopsy	Lead CpGs at *PTPN6/PHB2*, *ANKRD11*, and *TNRC18* map to active enhancers in kidney cortex (EWAS, fibrosis↑)
UUO and DKD kidney mice model, primary renal fibroblast [67]	Kidney of mice model, primary renal fibroblast	Hypermethylation of the *Rasal1* promoter (fibrosis↑)
High glucose treated human glomerular mesangial cells [65]	Mesangial cell	Reduced methylation of *CTGF* promoter (ECM↑)
Proximal tubules of *db/db* mice [53]	Tubular epithelia	Aberrant hypomethylation of *Agt, Abcc4, Cyp4a10, Glut5*, and *Met* and hypermethylation of *Kif20b, Cldn18, and Slco1a1* (mitochondria dysfunction)

Denote: T1D: Type 1 diabetic, EWAS: Epigenome-wide association studies, DM: Diabetes mellitus, DKD: Diabetic kidney disease, ECM: Extracellular matrix, EDIC: Epidemiology of Diabetes Interventions and Complications, and UUO: Unilateral ureteral obstruction. Symbols: ↑ refers to increase; ↓ refers to decrease.

**Table 2 ijms-23-00843-t002:** Histone modifications in diabetic kidney disease.

Study Design (Reference)	Main Cells or Tissue Samples	Epigenetic Changes (Mechanisms Involved)
30 DCCT conventional treatment subjects (cases: mean HbA1c level >9.1% with retinopathy or nephropathy by EDIC year 10 of follow-up) versus 30 intensive treatment subjects (controls: mean HbA1c level <7.3% without complications) [68]	Blood monocytes and lymphocytes	Promoter regions with enrichment H3K9Ac↑ (inflammation↑)
*db/db* mice [80]	Kidneys of *db/db* mice	SET7/9 and the recruitment to promoters↑, H3K4me1 recruitment at MCP-1 promoters↑ (ER stress↑)
Glomeruli from diabetic mice, TGF-β1 or high glucose treated rat mesangial cell [69]	Mesangial cell	H3K9/14 acetylation↑ and CBP/p300 occupancies↑ at the PAI-1 and P21 promoters (ECM↑)
Diabetic *db/db* mice, TGF-β treated mouse mesangial cell [70]	Mesangial cell	Akt and p300↑, acetylation of Ets-1 and histone H3↑ (ECM↑)
Type 1 diabetic model, high glucose- or sodium butyrate-treated mesangial cells in the presence or absence of apelin-13 [78]	Mesangial cell	Apelin-13 treatment inhibited histone hyperacetylation by upregulation of histone deacetylase (inflammation)
TGF-β1 treated rat mesangial cell under high or normal glucose [79]	Mesangial cell	SET7/9↑ at promoters of the ECM-associated genes (ECM↑)
High glucose treated mouse mesangial [81]	Mesangial cell	Suv39h1↓, H3K9me3 levels↓ at the promoters of fibronectin and p21(WAF1) genes (ECM↑)
Type 1 diabetic rat kidney [83]	Mesangial cell	H2AK119Ub↓ and H2BK120Ub↓ (ECM↑)
STZ-induced diabetic rats, TGF-β treated rat, mouse, and human mesangial cells [86]	Mesangial cell	miR-101b↑/Ezh2↓, Jmjd3 and Utx↑/H3K27me3↓ (mesangial dysfunction, ECM↑)
Kidney tissues from diabetic *db/db* mice and patients with DKD, high glucose-treated mouse podocytes [75]	Podocyte	HDAC9↑ (oxidative stress, apoptosis, and inflammation↑)
Human DKD renal biopsy, STZ-induced diabetic rats, diabetic *db/db* mice, glucose, or AGEs or TGF-β treated podocyte [76]	Podocyte	HDAC4↑ (inflammation↑)
Kidneys of diabetic rats, high glucose treated podocytes [84]	Podocyte	EZH2 expression↓ (oxidative stress↑)
Human FSGS or DKD renal biopsy, animal studies of adriamycin nephrotoxicity, subtotal nephrectomy and diabetic *db/db* mice, mouse and human podocytes [85]	Podocyte	H3K27me3↓ (podocyte dedifferentiation↑)
STZ-induced or obese-type (*db/db*) diabetic mice, high glucose treated human-derived renal epithelial cells [77]	Tubular epithelia	Sirt1↓ (podocyte foot process effacement↑)
Human DKD renal biopsy compared to non-DKD minimal change diseases, high glucose treated human proximal tubular epithelial cells [82]	Tubular epithelia	SUV39H1↑ (DM renal tubules), SUV39H1↓ (greater glucose and prolonged stimulation in cells) (inflammation↑)

Denote: DCCT: Diabetes Control and Complications Trial, EDIC: Epidemiology of Diabetes Interventions and Complications, ER: Endoplasmic reticulum, ECM: Extracellular matrix, STZ: Streptozotocin, DKD: Diabetic kidney disease, AGEs: Advanced glycation end products, and FSGS: Focal segmental glomerulosclerosis. Symbols: ↑ refers to increase; ↓ refers to decrease.

**Table 3 ijms-23-00843-t003:** Noncoding RNAs in diabetic kidney disease.

Study Design (Reference)	Main Cells or Tissue Samples	Epigenetic Changes (Mechanisms Involved)
MicroRNAs		
STZ-induced diabetes [88]	Kidney and macrophages of T1D mice	miR-146a↓ (inflammation↑)
Glomeruli of early DKD patients, renal cortex of diabetic (STZ-injected) mice [90]	Kidney of T1D mice	miR-192↑ (ECM, fibrosis↑)
PBMCs of DKD patients, kidneys of *db/db* mice, glucose treated mesangial cells [87]	Mesangial cell	miR-451↓ (inflammation↑)
STZ-injected diabetic mice, diabetic *db/db* mice, TGF-β treated mesangial cells [72]	Mesangial cell	miR-192↑ (ECM↑)
Glomeruli from diabetic (STZ-injected) mice, TGF-β treated glomerular mesangial cells [91]	Mesangial cell	miR-192↑ (ECM↑)
Type 2 diabetes in *db/db* mice, cultured mesangial cell [94]	Mesangial cell	miR-29b↓ (ECM, inflammation↑)
Glomeruli of diabetic mice, TGF-β treated mouse mesangial cells [102]	Mesangial cell	miR-200b/c↑ (mesangial hypertrophy, ECM↑)
Serum and kidney tissues of patients with DKD, *db/db* mice, cultured mesangial cells [103]	Mesangial cell	miR-135a↑ (ECM↑)
Mouse diabetic kidney disease models, high glucose or TGF-β treated human and mouse mesangial cells [104]	Mesangial cell	miR-377↑ (ECM↑)
Glomeruli of diabetic mice, TGF-β-treated mouse mesangial cells [107]	Mesangial cell	let-7 family members (let-7b/c/d/g/i) ↓ (ECM↑)
Glomeruli of diabetic patients, glomeruli of albuminuric BTBR *ob/ob* mice [89]	Podocyte	miR-146a↓ (podocyte injury↑)
Primary renal glomeruli from STZ-induced diabetic mice [93]	Podocyte	miR-29a↓ (podocyte injury↑)
Glomeruli of *db/db* mice, kidney microvascular endothelial cells and podocytes treated with high glucose [96]	Podocyte	miR-29c↑ (ECM↑)
High glucose treated mouse podocyte [97]	Podocyte	miR-34a↓ (podocyte apoptosis↑)
High glucose treated podocytes [98]	Podocyte	miR-34c↓ (podocyte apoptosis↑)
Humans and mouse (STZ-injected) models of DKD, cultured podocytes [105]	Podocyte	miR-26a↓ (ECM↑)
Renal biopsy from patients with diabetic kidney disease, high glucose treated proximal tubular cells [92]	Tubular epithelia	miR-192↓ (EMT↑)
Diabetic kidney disease animal models, cultured human tubular epithelial cells [100]	Tubular epithelia	miR-21↑ (ECM↑)
Human DKD, models of fibrotic renal disease and experimental DKD [101]	Tubular epithelia	miR-21↑ (ECM↑)
Mouse models of early and advanced DKD, TGF-β1 treated rat tubular cells [106]	Tubular epithelia	let-7b↓ (ECM↑)
Renal biopsies and plasma of DKD patients, STZ-induced diabetic rats, high glucose cultured rat proximal tubular cells [108]	Tubular epithelia	miR-130b↓ (EMT↑)
Early and advanced DKD mice models, TGF-β treated rat proximal-tubular cells [109]	Tubular epithelia	miR-141 and miR-200a ↓ (EMT↑)
Long noncoding RNAs		
Renal tissues of *db/db* DKD mice, mouse mesangial cell [99]	Mesangial cell	1700020I14Rik↓, miR-34a-5p↑(fibrosis↑)
Serum of DKD patients, mouse mesangial cells [110]	Mesangial cell	NR_033515↑, miR-743b-5p↓ (EMT↑)
Glomeruli of DKD mouse models, TGF-beta or high glucose treated mesangial cells [113]	Mesangial cell	lnc-MGC↑, a megacluster of microRNAs↑ (ER stress↑)
*db/db* mice, mouse mesangial cells and tubular epithelial cells [95]	Mesangial cell and tubular epithelia	Erbb4-IR↑, miR-29b↓(fibrosis↑)
Mesangial cells and human proximal tubular cells [114]	Mesangial cell and tubular epithelia	PVT1↑miR-1207-5p↑ (ECM↑)
Human DKD renal biopsy, diabetic rat, high glucose cultured podocyte [112]	Podocyte	LINC01619↓, miR-27a↑ (ER stress↑)
STZ-induced diabetic rats, high glucose treated renal tubular epithelial cell [111]	Tubular epithelia	MALAT1↑, miR-23c↓ (inflammation↑)
High glucose treated human proximal tubular cells [115]	Tubular epithelia	lncRNA MIR503HG↑, miR-503↑ (apoptosis↑)

Denote: STZ: Streptozotocin, T1D: Type 1 diabetic, DKD: Diabetic kidney disease, ECM: Extracellular matrix, PBMCs: Peripheral blood mononuclear cells, EMT: Epithelial-to-mesenchymal transition, and ER: Endoplasmic reticulum. Symbols: ↑ refers to increase; ↓ refers to decrease.

**Table 4 ijms-23-00843-t004:** Epigenetic biomarkers and agents to modulate epigenetics in diabetic kidney disease.

Epigenetic Biomarkers		
Status of DKD (Reference)	Samples	Epigenetic Biomarkers
DKD [118]	Whole blood	miR-126↓
DM with micro-/macrovascular complication [119]	Serum	miR-126↓
DM with microalbuminuria [116]	Urinary exosome	miR-192↑, miR-194↑, and miR-215↑
Type 1 DM with incipient diabetic kidney disease [117]	Urinary exosome	miR-130a↑, miR-145↑, miR-155↓, miR-424↓
DKD [120]	Urine	miR-2861↓, miR-1915-3p↓, miR-4532↓
Agents to Modulate Epigenetics		
Types of Nephropathy (Reference)	Agents to Modulate Epigenetics	Epigenetic Effects
DKD [122]	5-azacytidine or 5-aza-2’-deoxycytidine	DNA demethylation
DKD or CKD [67]	BMP7	Tet3↑ and normalization of *Rasal1* promoter hypermethylation
DKD [78]	Apelin-13	Histone deacetylation
DKD [123]	MS417	Bromodomain inhibitor

Denote: DKD: Diabetic kidney disease, DM: Diabetes mellitus, and CKD: Chronic kidney disease. Symbols: ↑ refers to increase; ↓ refers to decrease.

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
