# Peer review of "The Dynamics and Plasticity of Epigenetics in Diabetic Kidney Disease: Therapeutic Applications Vis-à-Vis"

_ijms, 2022, doi:10.3390/ijms23020843_

Round 1

Reviewer 1 Report

In this paper Kuo et al. present an excellent review of epigenetics in diabetic nephropathy. Physiology and pathophysiology for epigenetic mechanisms as well as for diabetic nephropathy very well explained and recent findings appropriately discussed. The work is well organized, with a multitude of references. The figures are well done. English language and stlye are fine. I recommend to publish the review in the current form.

Author Response

Thanks for the agreement and the kind affirmation of our manuscript.

Reviewer 2 Report

The manuscript is interesting and addresses a hot topic. Figures and tables are clear. The bibliography is up to date.

However, this reviewer raises some issues that need to be addressed by the authors.

1- From several years the definition of diabetic nephropathy (DN) has been replaced by diabetic kidney disease (DKD).DKD is defined as an increase in albuminuria and/or a reduction in GFR in the absence of other renal causes (Standard of Care. American Diabetes Association 2021). Therefore, the authors should replace the old with the new definition throughout the text.

2- Renal damage in diabetes is a typical expression of diabetic microangiopathy. Therefore, in addition to damage to mesangial cells, podocytes and tubular epithelia, endothelial damage plays a key role in DKD. Strangely, this issue has been omitted by the authors, while it deserves an extensive discussion in a dedicated chapter.

3- The authors in the abstract correctly point out that "The triggers for the irreversible renal function deterioration are multifactorial ...". However, in the text they do not underline the importance of a multifactorial approach of patients with DKD. Since primary and secondary prevention of morbidity and mortality (especially cardiovascular) is the challenge of modern therapy of DKD, it is appropriate to underline that very recently this goal has been reached for the first time by randomized multicenter NID study through a multifactorial intervention in albuminuric type 2 diabetic subjects (Cardiovasc Diabetol (2021) 20:145. doi: 10.1186/s12933-021-01343-1). This multifactorial strategy, in addition to the use of nephroprotective drugs such as SGLT2i, represents the future of the therapy of DKD patients. This issue and above reference should be discussed in conclusions.

4- The manuscript requires linguistic revision by a native English speaker.

Round 2

Reviewer 2 Report

The authors have addressed the issues raised by this reviewer, and the paper appears to be improved.
However, despite having accepted the observation that DKD is a myocroangiopathic complication of diabetes and inserted a (small) paragraph on epigenetic modifications of glomerular endothelial cells, both in the abstract and in the introduction the authors continue to indicate as "renal cell types crucial "only three cell types (mesangial cells, tubular epithelium and podocytes). Please, correct.

The paper still needs a linguistic review by a native English speaker

Author Response

Reviewer:

The authors have addressed the issues raised by this reviewer, and the paper appears to be improved.

However, despite having accepted the observation that DKD is a myocroangiopathic complication of diabetes and inserted a (small) paragraph on epigenetic modifications of glomerular endothelial cells, both in the abstract and in the introduction the authors continue to indicate as "renal cell types crucial "only three cell types (mesangial cells, tubular epithelium and podocytes). Please, correct.

The paper still needs a linguistic review by a native English speaker

Response : Accepted and revised

We have amended “three” crucial renal cell types to “several” and added glomerual endothelial cells in the abstract. We also delete the word “three” in the introduction.

The paper has been proceed through the English Editing.